# ViSTA: A Novel Network Improving Lung Adenocarcinoma Invasiveness Prediction from Follow-Up CT Series

**DOI:** 10.3390/cancers14153675

**Published:** 2022-07-28

**Authors:** Wei Zhao, Yingli Sun, Kaiming Kuang, Jiancheng Yang, Ge Li, Bingbing Ni, Yingjia Jiang, Bo Jiang, Jun Liu, Ming Li

**Affiliations:** 1Department of Radiology, The Second Xiangya Hospital, Central South University, Changsha 410011, China; wei.zhao@csu.edu.cn (W.Z.); moshangqingcheng@163.com (Y.J.); jiangbo@csu.edu.cn (B.J.); 2Department of Radiology, Huadong Hospital, Fudan University, Shanghai 200040, China; sunyingli208ok@126.com; 3Dianei Technology, Shanghai 200051, China; kaiming.kuang@dianei-ai.com (K.K.); jekyll4168@sjtu.edu.cn (J.Y.); 4Department of Electronic Engineering, Shanghai Jiao Tong University, Shanghai 200240, China; nibingbing@sjtu.edu.cn; 5Department of Radiology, The Xiangya Hospital, Central South University, Changsha 410008, China; ligeanyi@126.com; 6Radiology Quality Control Center, Changsha 410011, China; 7Institute of Functional and Molecular Medical Imaging, Fudan University, Shanghai 200437, China

**Keywords:** adenocarcinoma, invasiveness, X-ray computed tomography, deep learning, temporal attention

## Abstract

**Simple Summary:**

Assessing follow-up computed tomography(CT) series is of great importance in clinical practice for lung nodule diagnosis. Deep learning is a thriving data mining method in medical imaging and has obtained surprising results. However, previous studies mostly focused on the analysis of single static time points instead of the entire follow-up series and required regular intervals between CT examinations. In the current study, we propose a new deep learning framework, named ViSTA, that can better evaluate tumor invasiveness using irregularly serial follow-up CT images to avoid aggressive procedures or delay diagnosis in clinical practice. ViSTA provides a new solution for irregularly sampled data. ViSTA delivers superior performance compared with other static or serial deep learning models. The proposed ViSTA framework is capable of improving performance close to the human level in the prediction of invasiveness of lung adenocarcinoma while being transferrable to other tasks analyzing serial medical data.

**Abstract:**

To investigate the value of the deep learning method in predicting the invasiveness of early lung adenocarcinoma based on irregularly sampled follow-up computed tomography (CT) scans. In total, 351 nodules were enrolled in the study. A new deep learning network based on temporal attention, named Visual Simple Temporal Attention (ViSTA), was proposed to process irregularly sampled follow-up CT scans. We conducted substantial experiments to investigate the supplemental value in predicting the invasiveness using serial CTs. A test set composed of 69 lung nodules was reviewed by three radiologists. The performance of the model and radiologists were compared and analyzed. We also performed a visual investigation to explore the inherent growth pattern of the early adenocarcinomas. Among counterpart models, ViSTA showed the best performance (AUC: 86.4% vs. 60.6%, 75.9%, 66.9%, 73.9%, 76.5%, 78.3%). ViSTA also outperformed the model based on Volume Doubling Time (AUC: 60.6%). ViSTA scored higher than two junior radiologists (accuracy of 81.2% vs. 75.4% and 71.0%) and came close to the senior radiologist (85.5%). Our proposed model using irregularly sampled follow-up CT scans achieved promising accuracy in evaluating the invasiveness of the early stage lung adenocarcinoma. Its performance is comparable with senior experts and better than junior experts and traditional deep learning models. With further validation, it can potentially be applied in clinical practice.

## 1. Introduction

Low-dose computed tomography (LDCT) is recommended for lung cancer screening in high-risk populations based on the National Lung Cancer Screening Trial (NLST) report, which is now included in US screening guidelines [1]. Owing to LDCT, more and more early stage lung adenocarcinomas are diagnosed and treated. In clinical practice, most people require follow-up CT scans due to the indeterminate diagnosis or low probability of malignancy on baseline CT. Assessing the changes in size, CT value, and other imaging features can substantially help the diagnosis and invasiveness evaluation of early stage lung adenocarcinomas. However, the evaluation process is tedious and lacks objectivity, which means that radiologists could be overwhelmed by numerous serial CT image evaluations. Moreover, the features indicating malignancy may not be present in the early stages of lung adenocarcinoma. As we know, biological changes may precede morphological changes. Therefore, an efficient tool for objectively evaluating the changes and mining the internal patterns of lung nodules on serial CTs is of great importance.

Deep learning is a thriving data mining method in medical imaging and has obtained surprising results [2,3,4]. It can efficiently and automatically process medical images and has achieved promising performances on par with clinicians on various clinical tasks, including disease classification, medical image registration, and organ segmentation [5,6,7,8,9]. Previous studies have shown that deep learning could aid clinical decision-making for early lung cancer in disease management and invasiveness prediction [10,11,12,13,14]. However, most prior studies only included single-time CT scan images, while serial CT scan images were not fully investigated. Several powerful deep learning methods have been invented to process serial data, e.g., Long Short-term Memory, Gated Recurrent Unit Network, and Transformer [15,16,17]. Equipped with the aforementioned tools, a deep learning system can include serial images, better evaluate the biological behavior and changes, and then better predict different clinical events, such as prognosis, therapeutic effect, and subsequent growth patterns [18].

Serial deep learning models have achieved great success in serial data domains, including natural language processing, video classification, and speech recognition [15,19,20]. Nonetheless, it is important to notice that medical serial data such as electronic health records [21] or medical examinations are almost always sampled irregularly in time, separating them from the aforementioned modalities. Since the progression of the disease is strongly correlated with the time intervals between two time points, the asynchronous (sampled irregularly) nature of medical data requires special treatment. For example, by limiting sampling time intervals to 1, 3, and 6 months, deep learning methods proved effective in integrating multiple time points and improving the prediction of lung cancer treatment response [22]. However, this restriction on time intervals still limits the usage of the deep learning method in processing clinical serial data, epically for irregularly serial data.

In this article, we propose ViSTA (Visual Simple Temporal Attention), a deep learning framework capable of predicting the tumor invasiveness of pulmonary adenocarcinomas from Follow-up CT Series. The main contributions are three-fold: First, by introducing a simple temporal attention mechanism, we propose a new deep learning network, named ViSTA, to evaluate the invasiveness of early stage lung adenocarcinoma using irregularly serial CT scans images. ViSTA is able to gather information throughout the entire series and improve the prediction performance. Compared with serial analysis using traditional recurrent neural networks [22], ViSTA is not limited by different time intervals and can process completely irregularly sampled serial data. ViSTA was trained and validated on a dataset of 1121 CT scans from 282 follow-up series and evaluated on a hold-out test set of 113 CT scans from 69 follow-up series. Second, ViSTA delivers superior performances compared with other static or serial deep learning models. ViSTA also outperforms size-based predictive methods (Volume Doubling Time [23]) by a large margin. Third, ViSTA was proven to achieve higher scores than two junior radiologists and came close to one senior radiologist in the observer study. Our results prove ViSTA’s superiority in terms of processing irregularly sampled series and its great potential of being put into clinical practice in reality. Additionally, ViSTA is completely transferrable to other medical imaging tasks where analyzing serial data should yield better performances.

## 2. Materials and Methods

### 2.1. Data Collection

From January 2011 to October 2017, a search of the electronic medical records and the radiology information systems of the hospital was performed by one author (Yingli Sun). The inclusion criteria are as follows: (1) two or more available CT examinations with thin-slice (≤1.5 mm) images before resection. If there were only two CT examinations, the interval between two scans should be over 30 or more days. (2) Complete pathologic reports. The exclusion criteria for this analysis were: (1) prior treatment before surgery; (2) poor quality CT images; (3) lesions that were difficult to clearly delineate. Finally, a total of 351 nodules from 347 patients (mean age, 58.41 years ±11.79 (SD); range, 22–84 years) were enrolled in the study. Among the 351 lung nodules, 191 nodules were pathologically identified as preinvasive lesions, including 1 atypical adenomatous hyperplasia (AAH), 39 adenocarcinomas in situ (AIS), and 151 minimally invasive adenocarcinoma (MIA); whereas 160 nodules were identified as invasive adenocarcinoma (IA). In total, 1234 serials CT scans of the 351 nodules were enrolled in this study. The median interval between the first and the last CT examinations was 366 ± 500 days (range, 30–2813 days; interquartile range, 165–852 days). The 351 nodules were randomly separated into a training set (245 nodules), validation set (37 nodules), and test set (69 nodules) (see Table 1).

### 2.2. CT Scanning Parameters

Preoperative chest CT in our department was performed using the following four scanners: GE Discovery CT750 HD, 64-slice LightSpeed VCT (GE Medical Systems, Chicago, IL, USA); Somatom Definition flash, Somatom Sensation-16 (Siemens Medical Solutions, Erlangen, Germany) with the following parameters: 120 kVp; 100–200 mAs; pitch, 0.75–1.5; and collimation, 1–1.5 mm, respectively. All imaging data were reconstructed using a medium sharp reconstruction algorithm with a thickness of 1–1.5 mm.

### 2.3. Nodule Labeling, Segmentation and Imaging Preprocessing

A medical image processing and navigation software 3D Slicer (v4.8.0, Brigham and Women’s Hospital, Boston, MA, USA) was used to manually delineate the volume of interest (VOI) of the included nodules at the voxel level by one radiologist (Yingli Sun, with 5 years of experience in chest CT interpretation), then the VOI was confirmed by another radiologist (Ming Li, with 12 years of experience in chest CT interpretation). Large vessels and bronchioles were excluded as much as possible from the volume of the nodule. The lung CT DICOM (Digital Imaging and Communications in Medicine) format images were imported into the software for delineation, and then the images with VOI information were extracted with NII format for next step analysis. Each segmented nodule was attributed a specific pathological label (AAH, AIS, MIA, IA), according to the detailed pathological report. Two steps were performed to preprocess CT images before path extraction. First, the whole-volume CT image was resampled to the spacing of 1 mm in all three dimensions to guarantee isotropy. Second, HU values were clipped to the range of (−1000, 400) and normalized to (0, 1) using minimum–maximum normalization. Normalization can accelerate the convergence in the training of the deep learning model and improve its generalization ability.

### 2.4. Development of the Deep Learning Model

We developed a deep learning model named ViSTA to classify IA/non-IA lung nodules. The overall architecture of ViSTA is presented in Figure 1. ViSTA first extracts features from CT image patches using a CNN backbone and then integrates information from time series using a lightweight attention module named SimTA [24], which is designed specifically for analyzing asynchronous time series. Details regarding the architecture of ViSTA are provided in Appendix A, and a single SimTA layer was shown in Appendix A. To avoid overoptimization, we did not heavily tune the hyperparameters of our deep learning model and simply adopted common settings. ViSTA and all its counterparts are trained end-to-end for 100 epochs using the AdamW optimizer [25]. We used a cosine decay learning schedule from $10^{-3}$ to $10^{-6}$. The batch size of each update was 32. The drop-out probability and weight decay were set at 0.2 and 0.01 to avoid overfitting.

### 2.5. Counterpart Methods

For comparison with ViSTA, we conducted experiments on a few of its counterparts: –VDT (Volume Doubling Time). VDT is an important volumetric indicator used in follow-up examinations. It represents the time it takes for a nodule to double its volume. The formula of VDT is provided in Appendix A. Nodules with VDT < 400 days are considered fast-growing and are more likely to be malignant [23]. In this research, we evaluated VDT’s metrics under two different thresholds: 400 days and the cutoff that provides the best Youden index on the validation set. Youden index’s formula is presented in Appendix A;–CNN (Convolutional Neural Network): To compare ViSTA against static models, we introduced CNN as a counterpart. We conducted the following experiments to further investigate the source of performance difference between ViSTA and CNN;–CNN-last: This experiment was conducted to train and validate CNN only on the last time point of each follow-up series. It is obvious that the last time point is most relevant to the final diagnosis;–CNN-first: This experiment was conducted to train and validate CNN only on the first time point of each follow-up series. In this experiment, the first steps were treated as if they had the same label as the last one. This setting was used to confirm that earlier time points convey less information than later ones;–CNN-all: This experiment was conducted to train CNN on all time points of each follow-up series and validate it on the first and last time point separately (named CNN-all-first and CNN-all-last, respectively). This was used is to investigate if ViSTA’s superior performance only comes from the larger data size it enjoys;–CNN+LSTM (Long Short-term Memory) [16]: LSTM is a subtype of RNN (Recurrent Neural Network) designed to analyze serial data and capture long-term relations. This setting is quite similar to previous research which combined CNN and RNN to predict lung cancer treatment response [22]. However, we did not limit time intervals to specific values so that we could fairly compare ViSTA and RNN-based methods. One major difference between CNN+LSTM and ViSTA is that CNN+LSTM treats all time points as if they had the same interval (synchronous). By comparing the previous two methods, we would like to see if ViSTA is more suitable for analyzing irregularly sampled time series.

### 2.6. Evaluation and Statistical Analysis

We evaluated the proposed ViSTA model both quantitatively and qualitatively. To evaluate each method’s performance, we used a variety of metrics, including accuracy, precision, sensitivity, F1 score, and AUC. Formulas of evaluation metrics are presented in Appendix A.

To explore the visual representation and interpretability of ViSTA, we followed Simonyan, K. et al. [26] and plotted our model’s saliency maps through backpropagation, and investigated the mechanism under ViSTA and where it directed its attention.

### 2.7. Observer Study

To further evaluate the performance of ViSTA, we conducted an observer study to compare the performance of radiologists in the same task against other models. In the observer study, all 69 CT series in the test set were evaluated by three radiologists. One is a senior radiologist with 22 years of experience, and the other two are junior radiologists with 5 and 3 years of experience, respectively. Radiologists gave the results based on the evaluation of all available serials CTs. The reviewed results were analyzed and compared with the performance of our proposed model. Radiologists’ performances were evaluated using accuracy, sensitivity, precision, and F1 score.

## 3. Results

### 3.1. Performance of Deep Learning Models in Predicting the Invasiveness of Early Lung Adenocarcinoma

To validate the effectiveness of ViSTA in predicting IA/non-IA nodules, we evaluated its performance using a variety of metrics against its counterparts: VDT (cutoff value set at best Youden index or 400 days), CNN (including CNN-last, CNN-first, CNN-all-first and CNN-all-last), and CNN+LSTM.

Appendix A show their performances on the training dataset and validation dataset. Figure 2 provide the ROC curves of all models on the test dataset. Our proposed model outperformed all deep learning models and VDT-based methods in every metric by considerable margins (best among models are highlighted with an underscore). It is worth noting that VDT is far from effective in terms of invasiveness classification. It underperformed almost all deep learning models in terms of AUC, accuracy, and F1 score. Secondly, sequential models (ViSTA and CNN+LSTM) delivered better performances than CNN models that utilize static data points. ViSTA outperformed CNN+LSTM by considerable margins in all metrics. This performance gap can be attributed to ViSTA’s suitability to analyze asynchronous time series. Unlike CNN+LSTM, which treats all time points as if they were regularly sampled, ViSTA takes time intervals into account and is better at processing follow-up series. Furthermore, we trained CNN on all time points (CNN all-first and CNN all-last) to investigate if sequential models gain superiority over larger training datasets. It turned out that ViSTA and CNN+LSTM still outperformed CNN even when it was trained on all data.

### 3.2. Performance Comparison against Radiologists

In the observer study, we compared the performances of ViSTA and its counterparts against three radiologists (Table 2). One is a senior radiologist with 22 years of experience, and the other two are junior radiologists with 5 and 3 years of experience, respectively. All 69 follow-up series from the test set were included in the observer study. We evaluated radiologists’ performances using accuracy, sensitivity and precision, and F1 score and compared them against the proposed model. In terms of metrics that require specifying threshold, we chose the threshold that delivers the best Youden Index on the validation set as the cutoff value. Figure 2 plot deep learning models’ ROC curves against radiologists’ metrics. In terms of accuracy and F1 score, ViSTA scored higher than the two junior radiologists (accuracy of 81.2% vs. 75.4% and 71.0%; F1 score of 81.7% vs. 73.0% and 65.5%) and came close to the senior radiologist (accuracy of 81.2% vs. 85.5%; F1 score of 81.7% vs. 84.8%).

### 3.3. Visual Presentation Investigation

To investigate the mechanism of ViSTA, we used a neural network visualization technique [26] to visualize the attention heatmap of the model, which was mostly attributed to the predicted results and potentially correlated to the biological behavior (Figure 3). We took the absolute value of the raw heatmap and clipped it to the range of (0, 0.01) for better visualization and interpretation. In view of the created heatmaps, we can see that the “attention” of the deep learning system was mostly focused on the nodule. Areas surrounding the nodule draw the attention of ViSTA as well, meaning that they also carry valuable information as the nodule does (Figure 3A,B). Figure 3A show a long follow-up series of 11 time points. We observed that heatmaps stay blank in the first half of the series, during which both nodule volume and IA probability remain relatively stable. In the latter half, heatmaps begin to show along with significant increases in nodule volume and IA probability. Heatmaps are sometimes only lit up at the last time point (Figure 3B). We contribute this to the sudden increase of nodule volume between the third and the fourth time point, which provides sufficient information for the model. This argument is supported by the spike of IA probability at the fourth time point. In some rare cases, heatmaps on all time points are close to invisible (Figure 3C). We conjecture that this is because the lung nodule had almost no progression, which was proven by the fact that both nodule volume and IA probability stayed almost unchanged throughout the entire series.

## 4. Discussion

In the current study, we proposed a deep learning framework named ViSTA to predict the invasiveness of lung adenocarcinomas using serial CT images. Our results showed that models fed with serial CT images substantially and consistently outperformed models fed with single CT images. Moreover, our proposed model can effectively process asynchronous time series and outperform the traditional serial network, i.e., LSTM. Our models achieved an AUC of 86.4% and an F1 score of 81.7% in the test dataset, which were higher than those of all its counterparts. In the observer study, ViSTA achieved higher accuracy and F1 scores than two junior radiologists (accuracy of 81.2% vs. 75.4% and 71.0%, F1 score of 81.7% vs. 73.0% and 65.5%). When compared with the senior radiologist, our proposed model delivered close performance (accuracy of 81.2% vs. 85.5%, F1 score of 81.7% vs. 84.8%).

Timely and accurately assessing the biological behavior of early stage lung adenocarcinomas has been a continuous focus of attention in clinical practice. In contrast to traditional radiographic features and handcraft features, deeper and higher dimension level features mined by the deep learning method present promising advantages in many tasks, including predicting the invasiveness of the early lung adenocarcinoma. Kim et al. performed a comparison study and revealed that the predictive accuracy of the deep learning method was superior to those of the size-based logistic model [11]. We also analyzed the predictive value of VDT [27], a size-based key parameter in the differentiation of aggressive tumors from slow-growing tumors in clinical practice [24]. Not surprisingly, the performance of our proposed model substantially exceeded that of the VDT-based methods. It indirectly verified the conjecture that a deep learning system could extract and learn deeper and more valuable features, then better discover the biological behavior of the tumors and predict the invasiveness of early stage lung adenocarcinoma.

Although the deep learning method can obtain better performance, most previous studies only used single CT scan data prior to the surgery for training and extracting features, which cannot reveal and learn the internal growth pattern of the nodules. In clinical scenarios, internal growth is a vital component of Lung-RADS, a guideline to standardize image interpretation by radiologists and dictate management recommendations. Including serial CTs can facilitate medical tasks, such as differentiating benign tumors from malignant ones [28] and monitoring and predicting treatment response [22,29]. The discovery of our study supports this. By modelling serial CTs, the predictive performance of ViSTA substantially surpassed its counterparts in analyzing static data. In clinical practice, sequential medical data is generally sampled irregularly, i.e., with different follow-up time intervals. To address the irregular sampling issue, we adopted SimTA in our proposed model to process irregularly sampled time series. This lightweight module enables modeling sequential information in an efficient way. It turned out that the proposed ViSTA significantly outperformed the standard serial framework, i.e., CNN+LSTM, with considerably fewer parameters and less computation and memory footprint. ViSTA can better take advantage of the complete information of all time point CTs by modelling simple yet effective exponentially decay attention in time series. This was proved by our experiments comparing ViSTA, CNN+LSTM, and pure CNN models trained with all time point CTs (CNN-all). ViSTA’s superiority over CNN-all proved that its performance gain does not come from a larger training dataset.

In the visualization analysis, we found that ViSTA can drive its attention on the nodule and the surrounding tissue and drop more attention when the probability of invasiveness increases. It can partly explain the mechanism of the deep learning system. We also found some cases where the model appeared to use features close to the nodule, such as the vasculature and parenchyma surrounding the nodule. In fact, peritumoral tissue may possess valuable information, such as tumor-infiltrating status. Features extracted from the peritumoral tissues can improve the efficiency of intramodular radiomic analysis [30,31]. However, we still cannot fully interpret whether the model incorporates other abnormalities such as background emphysema in its predictions. Further investigation using more comprehensive model attribution techniques may allow clinicians to take advantage of the same visual features used by the model to assess the biological status of tumors. It is worth noting that some of the heatmaps in the time series are completely blue, meaning that the deep learning model allocated close to zero attention to these time points. Though this phenomenon is not completely interpretable, we argue that it can be attributed to these two facts: these time points are too far from the current one, and they lack findings informative for the deep learning model.

Even though the proposed ViSTA proved effective in processing irregularly sampled CT series in our experiments, there are several limitations left untouched. First, due to the difficulty of collecting complete lung nodule follow-up series, we only included data from a single center in this study. In clinical practice, it is preferable if the proposed method generalizes to multiple data domains. Furthermore, it is possible that a single follow-up series contains CT scans from different centers, which would be an important challenge to solve if the proposed model were to be put into clinical usage. In future studies, we will include CT series from external centers to validate the generalization performance of ViSTA. Second, the SimTA module in ViSTA models a simple temporal attention mechanism that monotonically increases weights as the time point gets closer to the current time. However, it is viable to model more complicated attention relations using deep learning models such as Transformeror Informer [15,32]. These temporal models enable capturing non-monotonic and dynamic temporal attention that could be useful in predicting invasiveness. Last but not least, even though we conducted a visual investigation on ViSTA, the interpretation of deep learning model predictions still remains a major challenge. Additionally, the final clinical decision is still up to clinicians to date. In our future research, we will further investigate the underlying mechanism of ViSTA or other similar attention mechanisms.

## 5. Conclusions

To summarize, we designed a deep learning model processing irregularly sampled CT series to predict the invasiveness of early stage lung adenocarcinoma from follow-up CT scans. The model achieved promising accuracy comparable with senior experts and better than junior experts and its counterparts. With further validation, the proposed model could better evaluate the invasiveness of early stage lung adenocarcinoma, avoiding aggressive procedures or delayed diagnosis and helping precise management in clinical practice.

## Figures and Tables

**Figure 1 cancers-14-03675-f001:**
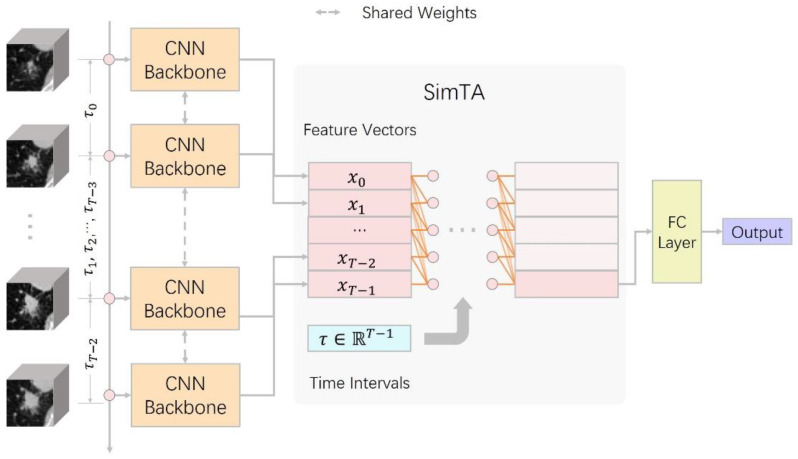
The model overview of the proposed ViSTA. It consists of a CNN backbone followed by the SimTA module made up of several SimTA layers.

**Figure 2 cancers-14-03675-f002:**
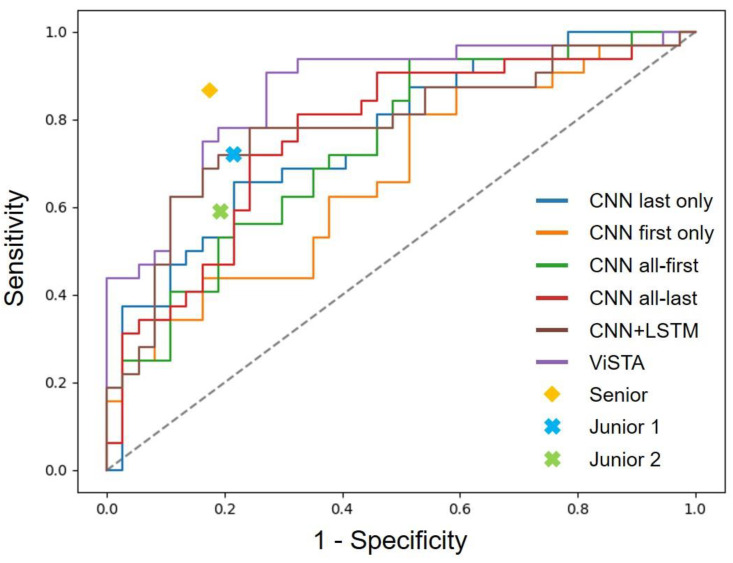
ROC curves of different models compared with performances of radiologists. The gray dotted line indicates the performance of a random classifier with no predictive ability.

**Figure 3 cancers-14-03675-f003:**
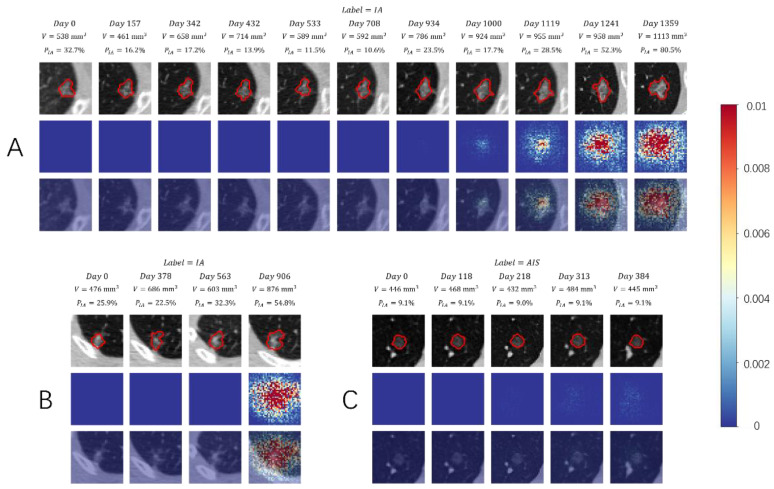
Visualization investigation of ViSTA. The top row shows CT slices of each time point in the follow-up series. The middle row shows attention heatmaps extracted using the technique proposed by Simonyan, K. et al. [26]. The bottom row masks heatmaps on top of CT slices. (**A**) Attention gradually grew along with the nodule volume and IA probability as the nodule progressed to the end of the series. (**B**) The heatmap only lit up at the last time point as it is considered the one carrying valuable information. (**C**) All time points are allocated with little to no attention, which may be caused by the slow progress of the nodule.

**Table 1 cancers-14-03675-t001:** Number of CT scans/nodules in training, validation, and test set.

Pathological Type		No. CT Scans/Nodules			
		Training	Validation	Test	Total
Non-IA	AAH	5/1	0/0	0/0	5/1
	AIS	98/29	9/4	19/6	126/39
	MIA	383/104	40/16	114/31	537/151
	Total	486/134	49/20	133/37	668/191
IA		398/111	64/17	104/32	566/160
Total		884/245	113/37	237/69	1234/351

**Table 2 cancers-14-03675-t002:** The performance of different models and radiologists on the test dataset. The highest among all is highlighted in bold, and the highest among models and VDT (Volume Doubling Time)-based methods is highlighted with an underscore.

	AUC	Acc.	Prec.	Sens.	F1
Senior	-	**85.5%**	**82.4%**	87.5%	**84.8%**
Junior 1	-	75.4%	74.2%	71.9%	73.0%
Junior 2	-	71.0%	73.1%	59.4%	65.5%
1/VDT (best Youden index)	60.6%	62.3%	56.3%	84.4%	67.5%
1/VDT (400 days)	60.6%	58.0%	71.4%	15.6%	25.6%
CNN last only	75.9%	72.5%	72.4%	65.6%	68.9%
CNN first only	66.9%	65.2%	70.0%	43.8%	64.3%
CNN all-first	73.9%	65.2%	60.5%	71.9%	65.7%
CNN all-last	76.5%	73.9%	71.9%	71.9%	71.9%
CNN+LSTM	78.3%	76.8%	73.5%	78.1%	75.8%
ViSTA	** 86.4% **	81.2%	74.4%	** 90.6% **	81.7%

## Data Availability

Data are available for bona fide researchers who request it from the authors.

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
