# Peer review of "ViSTA: A Novel Network Improving Lung Adenocarcinoma Invasiveness Prediction from Follow-Up CT Series"

_cancers, 2022, doi:10.3390/cancers14153675_

Round 1

Reviewer 1 Report

More reference about the topic such as

Assessing invasiveness of subsolid lung adenocarcinomas with combined attenuation and geometric feature models by Constance de Margerie‑Mellon et al

or 

Predicting the invasiveness of lung adenocarcinomas appearing as ground-glass nodule on CT scan using multi-task learning and deep radiomics by Xiang Wang et al

Author Response

Comments and Suggestions for Authors

More reference about the topic such as Assessing invasiveness of subsolid lung adenocarcinomas with combined attenuation and geometric feature models by Constance de Margerie‑Mellon et al or  Predicting the invasiveness of lung adenocarcinomas appearing as ground-glass nodule on CT scan using multi-task learning and deep radiomics by Xiang Wang et al

Answer: Thanks for your valuable suggestion. We have added them into our manuscript (Ref 13 and 14). It really improves the scientificity and comprehensiveness of our manuscript.

Reviewer 2 Report

In this paper, authors proposed a new deep learning network based on temporal attention, named ViSTA, to process irregular sampled follow-up CT scans and conducted substantial experiments to investigate the supplemental value in predicting the invasiveness using serial CTs. The following review comments are recommended, and the authors are invited to explain and modify.

Comment: Novelty is confusing. A highlight is required. The main contributions of the manuscript are not clear. The main contributions of the ‎article must be very clear and would be better if summarize ‎them into 3-4 points at the ‎end of the introduction.‎

Comment: “ViSTA first extracts features from CT image patches”, how to extract image patches? Whether these patches are overlapping or non-overlapping and what’s sizes?

Comment: How to preprocess CT images before image patch extraction?

Comment: Nothing is mentioned about the implementation challenges.

Comment: How to optimize hyper parameters during training?

Comment: Discuss the stability of the system in terms of complexity.

Comment: The following clinical decision support systems using AI (DL), and medical imaging must be included to improve the quality of the paper.

·       10.1155/2022/2665283

·       10.3390/math10050796

Comment: Moreover, it should be noticed that the clinical appliance has to be decided by medicals since the existing differences between the real image and the one generated by the proposed system could be substantial in the medical field.

Comment: Could you please check your references carefully ? All references must be complete before the acceptance of a manuscript.

Author Response

Comments and Suggestions for Authors

In this paper, authors proposed a new deep learning network based on temporal attention, named ViSTA, to process irregular sampled follow-up CT scans and conducted substantial experiments to investigate the supplemental value in predicting the invasiveness using serial CTs. The following review comments are recommended, and the authors are invited to explain and modify.

  1. Comment: Novelty is confusing. A highlight is required. The main contributions of the manuscript are not clear. The main contributions of the article must be very clear and would be better if summarize them into 3-4 points at the ‎end of the introduction.‎

Answer: Thanks for your valuable suggestion. It really improves our manuscript a lot. We have rearranged the sentences and summarized 3 points of main contributions in the Introduction section. Please see the revised version.

  1. Comment: “ViSTA first extracts features from CT image patches”, how to extract image patches? Whether these patches are overlapping or non-overlapping and what’s sizes?

Answer: Thanks for your professional comments. In this article, image patches were cropped around the center of pulmonary nodules, whose locations are labeled by experienced radiologists in advance. Please note that our research did not include pulmonary nodule detection. Our proposed method diagnoses nodules given their locations. Therefore, the concept of overlapping/non-overlapping patches/windows did not apply to our research.

  1. Comment: How to preprocess CT images before image patch extraction?

Answer: Thanks for your professional comment. There are two steps in our preprocessing. First, the whole-volume CT image is resampled to the spacing of 1mm in all three dimensions to guarantee isotropy. Second, HU values are clipped to the range of [-1000, 400] and normalized to [0, 1] using minimum-maximum normalization. Normalization accelerates the convergence in the training of the deep learning model and improves its generalization ability. We have added the information in the revised manuscript. (line 163-168, Page 4).

  1. Comment: Nothing is mentioned about the implementation challenges.

Answer: Thanks for your professional comment. It is indeed necessary to mention the implementation challenges in our manuscript. However, the main focus of our research is to facilitate clinical diagnosis using the deep learning technique. That is why we did not bring up the implementation details that much in our previous manuscript. Given your valuable comments, we have added implementation details in revised manuscript (line 175-180, Page 4).

  1. Comment: How to optimize hyper parameters during training?

Answer: Thanks for your professional comment. We did not heavily tune the hyperparameters of our deep learning model. We used common settings for hyperparameters such as model capacity, drop-out probability, regularization strength, and learning rate schedule. We believe that over-optimization of our hyperparameters might injure the generalization performance of the proposed framework.

Thank you for the suggestion. We have added the information required as explained above (line 175-180, Page 4).

  1. Comment: Discuss the stability of the system in terms of complexity.

Answer:

  • Most deep learning models may be over-parameterized, or overly complex. In order to improve the stability of the proposed model, we deployed weight decay and drop-out in training the proposed model. Both of these two tricks are commonly used in the field of deep learning to reduce overfitting or instability. We did not further investigate the relation between stability and model complexity, as this is not within the concern of this research. Our main focus lies in improving diagnosis accuracy using the deep learning technique. Nonetheless, we have added our implementation details of reducing model instability and overfitting to revised manuscript.(line 175-180, Page 4)
  1. Comment: The following clinical decision support systems using AI (DL), and medical imaging must be included to improve the quality of the paper.1155/2022/2665283; 10.3390/math10050796

Answer: Thanks for your valuable suggestion. We have added the above two valuable papers as reference in our manuscript.

  1. Comment: Moreover, it should be noticed that the clinical appliance has to be decided by medicals since the existing differences between the real image and the one generated by the proposed system could be substantial in the medical field.

Answer: Thanks for your professional comment. You must be an excellent expert in the area of smart medical research. Yes, you are right that the clinical appliance has to be decided by medicals. However, firstly we need excellent experts, like you, to construct perfect deep learning models to help clinical decision making. Generally, we often take two strategies to improve the performance of the model. One is using high-quality images or data that best fit the real world. Another is to design a suitable network to tackle the clinical issue. In clinical practice, the best working pattern is that the proposed deep learning models provide important information for clinical issues, and then the clinicians make their final decision partly upon the results provided by deep learning models. With the development of deep learning and efforts made by experts, like you, I think clinicians may more rely on the results provided by models. In our manuscript, the proposed model actually does not generate images. We only analyze the real CT images to give the invasiveness extent of lung adenocarcinoma, then to avoid aggressive procedures or delay diagnose in clinical practice. Of course, as you said, the final decision is up to clinicians. We have added the statement in our revised manuscript (limitation section).

  1. Comment: Could you please check your references carefully ? All references must be complete before the acceptance of a manuscript.

 Answer: Thank you for underlining this deficiency. We have checked all references to make sure all are relevant to the contents of the Manuscript.

Reviewer 3 Report

The authors present a new deep learning (DP) convolutional neural network, ViSTA, to predict lung adenocarcinoma invasiveness from computed tomography (CT) scans. A great advantage of ViSTA over other DP frameworks is its ability to process CT scans sampled irregularly over time, as is the case in medical visits and follow-ups. The authors’ data show that ViSTA performs better than other preexisting DP frameworks and comes close to radiologists’ diagnoses –radiologists with both intermediate and long experience were employed in this study. The authors appropriately present the background, materials, and methods of their study. They also appropriately discuss and sustain their main results, although their sample for testing, prediction and validation might be somewhat small. Many relevant mathematical formulas are presented and explained in the supplementary materials.

I think that this manuscript presents an interesting and promisingly accurate DP approach for lung cancer diagnosis. Therefore, I recommend this manuscript to be published in Cancers once the authors satisfactorily address the following secondary issues:

1. The written English of this manuscript is satisfactory for a scientific communication. However, the authors can still improve the English fluency and split some long sentences into two.

2. The authors should correct a few minor grammatical mistakes (for instance, on line 24, “can better evaluate…” should be “that can evaluate better…”) and some incorrectly split words (like “fo ur” on lines 132-133).

3. Starting on line 190, most of the text is oddly written in italics: this should be corrected.

4. The supplementary materials are called “supplementary methods” in the text: is this correct?

5. In the supplementary materials, the same figure is presented twice with and without a caption: is this correct?

Author Response

Comments and Suggestions for Authors

The authors present a new deep learning (DP) convolutional neural network, ViSTA, to predict lung adenocarcinoma invasiveness from computed tomography (CT) scans. A great advantage of ViSTA over other DP frameworks is its ability to process CT scans sampled irregularly over time, as is the case in medical visits and follow-ups. The authors’ data show that ViSTA performs better than other preexisting DP frameworks and comes close to radiologists’ diagnoses –radiologists with both intermediate and long experience were employed in this study. The authors appropriately present the background, materials, and methods of their study. They also appropriately discuss and sustain their main results, although their sample for testing, prediction and validation might be somewhat small. Many relevant mathematical formulas are presented and explained in the supplementary materials.

I think that this manuscript presents an interesting and promisingly accurate DP approach for lung cancer diagnosis. Therefore, I recommend this manuscript to be published in Cancers once the authors satisfactorily address the following secondary issues:

  1. The written English of this manuscript is satisfactory for a scientific communication. However, the authors can still improve the English fluency and split some long sentences into two.

Answer:Thanks for your valuable suggestion. We have modified the written English according to your suggestion. Please tell us if the current version is still need improved. Thanks.

  1. The authors should correct a few minor grammatical mistakes (for instance, on line 24, “can better evaluate…” should be “that can evaluate better…”) and some incorrectly split words (like “fo ur” on lines 132-133).

Answer:Thanks for your kindly reminder and we are sorry for the mistakes. We have rephrased the sentence “can better evaluate…” and recorrected the split errors according to the comment. We also checked throughout the text for similar mistakes.

  1. Starting on line 190, most of the text is oddly written in italics: this should be corrected.

Answer:We are grateful for the suggestion. We have modified the format of the corresponding section.

  1. The supplementary materials are called “supplementary methods” in the text: is this correct?

Answer:Thank you for underlining the inappropriate description. We have modified it into supplemental material.

  1. In the supplementary materials, the same figure is presented twice with and without a caption: is this correct?

Answer:Thanks for your kindly reminder and we are sorry for the mistakes. We have deleted the duplicated figure and added the caption for the figure. Thanks again.

Round 2

Reviewer 2 Report

The authors have answered my questions satisfactorily and I have no more comment.